# Candidate Set Re-ranking for Composed Image Retrieval with Dual Multi-modal Encoder

**Zheyuan Liu**                                                                 *zheyuan.liu@anu.edu.au*
*Australian National University*

**Weixuan Sun**                                                                 *weixuan.sun@anu.edu.au*
*Australian National University*

**Damien Teney**                                                                 *damien.teney@idiap.ch*
*Idiap Research Institute*
*Australian Institute for Machine Learning (AIML)*

**Stephen Gould**                                                               *stephen.gould@anu.edu.au*
*Australian National University*

**Reviewed on OpenReview:** *https://openreview.net/forum?id=fJAwemcvpL*

## Abstract

Composed image retrieval aims to find an image that best matches a given multi-modal user query consisting of a reference image and text pair. Existing methods commonly pre-compute image embeddings over the entire corpus and compare these to a reference image embedding modified by the query text at test time. Such a pipeline is very efficient at test time since fast vector distances can be used to evaluate candidates, but modifying the reference image embedding guided only by a short textual description can be difficult, especially independent of potential candidates. An alternative approach is to allow interactions between the query and every possible candidate, i.e., reference-text-candidate triplets, and pick the best from the entire set. Though this approach is more discriminative, for large-scale datasets the computational cost is prohibitive since pre-computation of candidate embeddings is no longer possible. We propose to combine the merits of both schemes using a two-stage model. Our first stage adopts the conventional vector distancing metric and performs a fast pruning among candidates. Meanwhile, our second stage employs a dual-encoder architecture, which effectively attends to the input triplet of reference-text-candidate and re-ranks the candidates. Both stages utilize a vision-and-language pre-trained network, which has proven beneficial for various downstream tasks. Our method consistently outperforms state-of-the-art approaches on standard benchmarks for the task. Our implementation is available at
https://github.com/Cuberick-Orion/Candidate-Reranking-CIR.

## 1 Introduction

The task of composed image retrieval (CIR) aims at finding a candidate image from a large corpus that best matches a user query, which is comprised of a reference image and a modification sentence describing certain changes. Compared to conventional image retrieval setups such as text-based (Li et al., 2011) or content-based (Tong & Chang, 2001) retrieval, the incorporation of both the visual and textual modalities enables users to more expressively convey the desired concepts, which is useful for both specialized domains such as fashion recommendations (Wu et al., 2021; Han et al., 2017) and the more general case of searching over open-domain images (Liu et al., 2021; Couairon et al., 2022; Delmas et al., 2022).

Existing work (Vo et al., 2019; Dodds et al., 2020; Chen et al., 2020a; Baldrati et al., 2022a) on CIR mostly adopts the paradigm of separately embedding the input visual and textual modalities, followed by a model

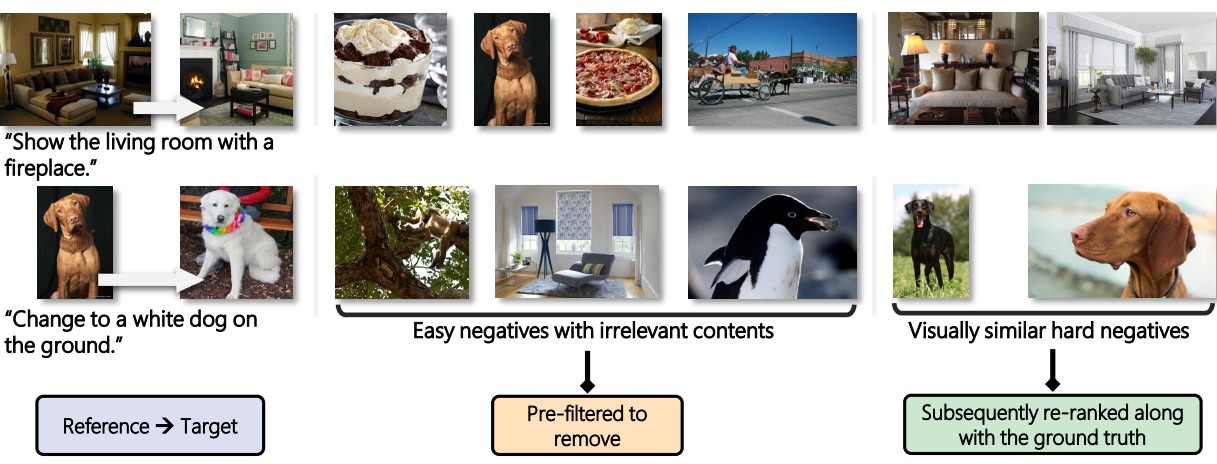

Figure 1: An illustration of the two-stage scheme, with easy negatives pre-filtered out, and the remaining candidates re-ranked.

that acts as an image feature modifier conditioned on the text. The modified image feature is finally compared against features of all candidate images through vector distances (i.e., cosine similarities) before yielding the most similar one as the prediction (see Figure 2 left). The main benefit of such a pipeline is the inference cost. For a query consisting of a reference image $I_R$ and a modification text $t$, to select the best matching candidate, the model shall exhaustively assess triplets $\langle I_R, t, I_C \rangle$ for every image $I_C \in \mathcal{D}$, where $\mathcal{D}$ is the candidate image corpus. Such an exhaustive pairing of $\langle I_R, t \rangle$ and $I_C$ results in a large number of query-candidate pairs. In the pipeline discussed above, all candidate images are individually pre-embedded, and the comparison with the joint-embedding of $\langle I_R, t \rangle$ is through the cosine similarity function that is efficient to compute even at a large scale. We point out that such a pipeline presents a trade-off between the inference cost and the ability to exercise explicit query-candidate reasoning. In essence, the candidate images are only presented to the model indirectly through computing the cosine similarity and the loss function, resulting in the model having to estimate the modified visual features from text inputs in its forward path.

To this end, we propose a solution that exhaustively classifies the $\langle I_R, t, I_C \rangle$ triplets, which enables richer query-candidate interactions — thus, achieving an appreciable performance gain — while still maintaining a reasonable inference cost. We observe that for CIR, easy and hard negatives can be distinctly separated, as the nature of this task dictates that the ground truth candidate be visually similar to the reference, otherwise it would be trivial to study a modification (Liu et al., 2021; Couairon et al., 2022). We then further deduce that a group of hard negatives exist, which is likely to benefit from fine-grained multi-modal reasoning, as illustrated in Figure 1. This motivates a two-stage method, where we first filter all candidates to reduce their quantity. Since the goal at this stage is to remove easy negatives, a low-cost vector distance (i.e., cosine similarity)-based pipeline would suffice. We then re-rank the remaining candidates with explicit text-image matching on each possible triplet. Granted, such a process is more computationally intense but is empirically beneficial for reasoning among hard candidates. With the pre-filtering in place, we are able to limit the overall inference time within an acceptable range. The main focus of this paper is on the second stage.

Note that our two-stage pipeline relates to the inference scheme of image-text retrieval (Lin et al., 2014) in recent vision-and-language pretrained (VLP) networks (Li et al., 2021; 2022). Specifically, Li et al. (2021) propose to first compute feature similarities for all image-text pairs, then re-rank the top-$k$ candidates through a joint image-text encoder via the image-text matching (ITM) scores, which greatly speeds up the inference compared to previous VLP networks that require computing ITM scores for *all* image-text pairs (Chen et al., 2020b; Li et al., 2020). Here, we arrive at a similar two-stage scheme but for the task of CIR. We also note that our method, although sharing a similar philosophy and is based on VLP networks, is not a direct replica of what is done in the image-text retrieval tasks discussed above. With the unique input triplets of $\langle I_R, t, I_C \rangle$, novel model architectures are required for efficient interactions among the three features of two modalities.

In summary, our contribution is a two-stage method that combines the efficiency of the existing pipeline and the ability to assess fine-grained query-candidate interactions through explicit pairing. We base our design on VLP models while developing task-specific architectures that encourage interactions among input entities. Our approach significantly outperforms existing methods on datasets of multiple domains.

## 2 Related Work

The task of image retrieval traditionally accepts input in the form of either an image (Tong & Chang, 2001) or text (Zhang et al., 2005; Li et al., 2011). The aim is to retrieve an image whose content is the most similar to the input one, or respectively, best matches the textual description. Vo et al. (2019) propose composed image retrieval (CIR), which takes as input both modalities, using an image as a reference while text as a modifier.

Current approaches address this task by designing models that serve as a reference image modifier conditioned on text, essentially composing the input modalities into one joint representation, which is compared with features of candidates through cosine similarity. Among them, TIRG (Vo et al., 2019) uses a gating mechanism along with a residual connection that aims at finding relevant changes and preserving necessary information within the reference respectively. The outputs of the two paths are summed together to produce the final representation. Anwaar et al. (2021) follow a similar design but pre-encode the inputs separately and project them into a common embedding space for manipulation. Hosseinzadeh & Wang (2020) propose to adopt regional features as in visual question answering (VQA) (Anderson et al., 2018) instead of convolutional features. Likewise, Wen et al. (2021) develop global and local composition networks to better fuse the modalities. VAL (Chen et al., 2020a) introduces a transformer network to jointly encode the input modalities, where the hierarchical design encourages multi-layer matching. MAAF (Dodds et al., 2020) adopts the transformer network differently by pre-processing the input into sequences of tokens to be concatenated and jointly attended. Yang et al. (2021) designs a joint prediction module on top of VAL that highlights the correspondences between reference and candidate images. Notably, the module is only used in training as it is intractable to apply it to every possible pair of reference and candidate images during inference. CIRPLANT (Liu et al., 2021) proposes to use a pre-trained vision-and-language (VLP) transformer to modify the visual content, alongside CLIP4CIR (Baldrati et al., 2022a;b), BLIP4CIR (Liu et al., 2024) and CASE (Levy et al., 2023).

DCNet (Kim et al., 2021) introduces the composition and correction networks, with the latter accepting a reference image with a candidate target image and assessing their relevancy. This, on first look, suggests an exhaustive reference-candidate pairing. Though, inference cost limits the interaction of a pair of reference and candidate images to simple operations — i.e., element-wise product and subtraction with a single-layer multi-layer perceptron (MLP). ARTEMIS (Delmas et al., 2022) is the first to introduce a model that scores each pair of query and candidate image, which separates it apart from an image modifier-based pipeline. However, inference cost still confines such scoring to cosine similarities between individually pre-encoded modalities. In contrast to existing approaches, our method is in two stages. We do not seek to modify image features in an end-to-end manner. Instead, we pre-filter the candidates and focus more on re-ranking the remaining hard negatives. The re-ranking step is formatted as a scoring task based on contrastive learning, which is natural for VLP networks trained with similar objectives.

We note that the concept of a two-stage scheme is not new for conventional image-text or document retrieval. Indeed, re-ranking a selected list of candidate images via e.g., k-nearest neighbors (Shen et al., 2012) or query expansion techniques (Chum et al., 2007) has been widely studied. More recent and related work on VLP models (Li et al., 2021; 2022; 2023) propose to first score the similarities between image and text features, then re-rank the top-$k$ pairs via a multi-modal classifier. This aligns nicely with the two pre-training objectives, namely, image-text contrastive and image-text matching. To the best of our knowledge, we are the first to apply such a two-stage scheme to CIR. We contribute by designing an architecture that reasons over the triplet of $\langle I_{\mathrm{R}}, t, I_{\mathrm{C}} \rangle$, which differs from the conventional retrieval tasks discussed above.

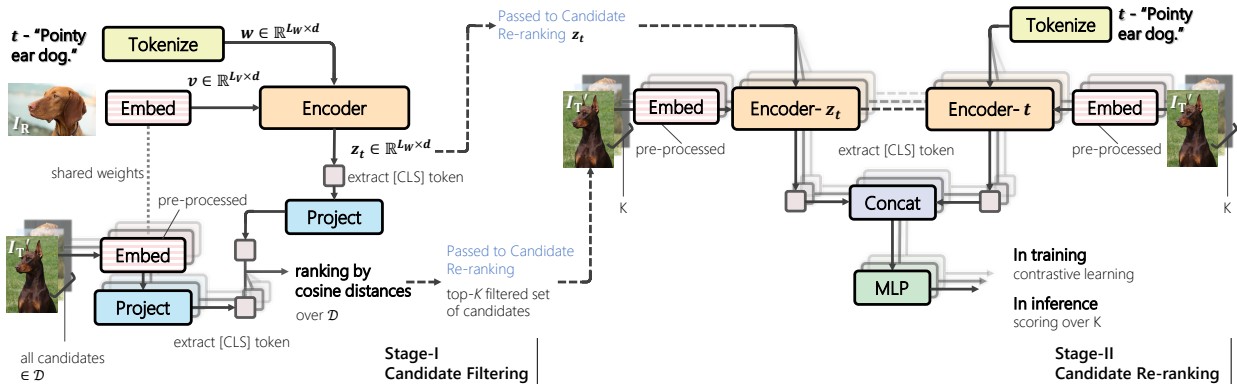

Figure 2: Overall training pipeline. In both stages, we freeze the image encoders (dashed fillings), as detailed in Section 4.1.3. **(Left)** Candidate filtering model, which takes as input the tokenized text and cross-attends it with the reference image. The output is the sequential feature $z_t$, where we extract the `[CLS]` token as the summarized representation of the query $q = \langle I_R, t \rangle$ to compare its similarity with features of $I'_T$. **(Right)** Candidate re-ranking model with dual-encoder architecture. Stacked elements signify that we exhaustively pair up each candidate $I'_T$ among the selected top-$K$ with the query $q$ for assessment. Note that the two encoders take in different inputs for cross-attention. The output `[CLS]` tokens are concatenated and passed for producing a logit. Note that the two stages are two separate models and not jointly trained.

## 3  Two-stage Composed Image Retrieval

The task of CIR can be defined as follows. Let $I_R$ be some reference image and $t$ be a piece of text describing a change to the image. Then given a query consisting of the pair $q = \langle I_R, t \rangle$, the aim of CIR is to find the best match, i.e., the target image $I_T$, among all candidates in a large image corpus $\mathcal{D}$. In this work, we propose a two-stage model where we first filter the large corpus to obtain a smaller set of candidate images relevant to the query (see Section 3.1), and then re-rank to obtain an ordered list of target images (see Section 3.2).

For both steps, we base our designs on the vision-and-language pre-trained (VLP) network BLIP (Li et al., 2022), though other VLP models might be used. BLIP consists of an image encoder and a text encoder. The image encoder is a vision transformer (Dosovitskiy et al., 2021) that accepts as input a raw image and produces the spatial image features by slicing the image in patches and flattening them in a sequence. A global image feature is also represented as a prepended special `[CLS]` token. The text encoder can operate in three modes. When configured as a uni-modal encoder, it takes in a sequence of tokenized words from a text sequence and outputs the sequential features with a `[CLS]` token summarizing the whole text, as in BERT (Devlin et al., 2019). Optionally, the text encoder can be configured as a multi-modal encoder, where a cross-attention (CA) layer is inserted after each self-attention (SA) layer. As shown in Figure 3, the CA layer accepts the sequential output of the image encoder and performs image-text attention. The output of which is passed into the feed-forward (FF) layer and is the same length as the input text sequence. The transformer-based text encoder accepts inputs of varied lengths while sharing the same token dimension $d$ as the output of the image encoder. In this paper, we denote the features of an arbitrary image (resp. input text) as $\boldsymbol{v}$ (resp. $\boldsymbol{w}$) and its length as $L_{\boldsymbol{v}}$ (resp. $L_{\boldsymbol{w}}$). We note that a decoder mode is also available in BLIP for generative tasks (e.g., image captioning (Anderson et al., 2018)), though it is not used in this work.

### 3.1  Candidate Filtering

The first stage of our approach aims to filter out the majority of candidates leaving only a few of the more difficult candidates for further analysis in the second step. Shown in Figure 2 (left), we adopt the BLIP text encoder in its multi-modal mode such that it jointly embeds a given query $q = \langle I_R, t \rangle$ into a sequential output, which we denote as $z_t \in \mathbb{R}^{L_{\boldsymbol{w}} \times d}$. Note that the output sequence $z_t$ is of length $L_{\boldsymbol{w}}$ as the input text, a characteristic that is further exploited in the second stage model. We extract the feature of the `[CLS]` token in $z_t$ as a single $d$-dimensional vector and compare it to pre-computed `[CLS]` embeddings of all candidate

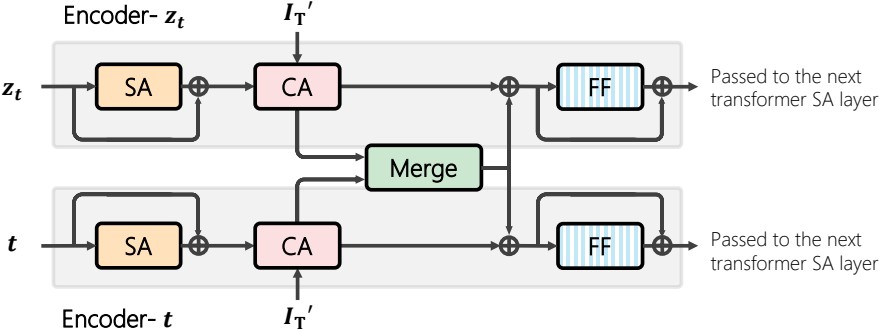

Figure 3: Details of the transformer layer in our dual-encoder architecture. Here, we take the first layer as an example. **SA**: Self-attention layer, **CA**: Cross-attention layer, **FF**: Feed-forward layer. $\oplus$: element-wise addition for residual connections. All modules in the figure are being trained. Dashed fillings on FF suggest weight-sharing.

images $I'_T \in \mathcal{D}$ via cosine similarity. Here, the pre-computed candidate embeddings are independent of the query text, which is a weakness that we address in our second stage model. Since BLIP by default projects the `[CLS]` token features into $d = 256$ for both image and text, the comparison can be efficiently done through the cosine similarity function.

After training the candidate filtering model, we select the top-$K$ candidates (for each query) for re-ranking in the second stage. Here we choose $K$ to be sufficiently large so that the ground-truth target is empirically observed to be within the selected images for most queries. We term the percentage value of queries with ground truth within the top-$K$ as *ground truth coverage* in the following sections. Empirically, we find that setting $K$ to 50 or 100 gives a good trade-off between recall and inference cost in the second stage. Details on the ablation of $K$ across datasets are discussed in Section 4.3.2.

We note that concurrent to our work, CASE (Levy et al., 2023) adopts a similar approach as our candidate filtering model, in that it uses BLIP for reference image-text fusion. We point out that both our filtering model and CASE use BLIP in one of its originally proposed configurations without architectural changes, and hence, is unsurprising and could be viewed as a natural progression for the task. Meanwhile, our second-stage re-ranking sets us apart from this concurrent work.

## 3.2   Candidate Re-ranking

The second stage re-ranks the filtered set of candidates. Since this set is much smaller than the entire corpus we can afford a richer, more expensive approach. To this end, we introduce a novel dual-encoder design for this subtask inspired by the BLIP architecture proposed for NLVR (Suhr et al., 2019).

As shown in Figure 2 (right), our two encoders run in parallel as two branches to serve separate purposes, one to encode $t$ with $I'_T$ (Encoder-$t$) and the other to encode $I_R$ with $I'_T$ (Encoder-$z_t$). Internally, they exchange information via dedicated merging layers. Specifically, Encoder-$t$ takes as input the tokenized $t$, which is then embedded into a sequential feature of size $\mathbb{R}^{L_w \times d}$. Meanwhile, Encoder-$z_t$ accepts as input $z_t \in \mathbb{R}^{L_w \times d}$ from the previous stage, which is a surrogate of $I_R$ (further discussed below). Since models of the two stages are trained separately, here we can pre-compute $z_t$ for each query of $q = \langle I_R, t \rangle$ using the first-stage candidate filtering model. We note that for a given query, the lengths of $z_t$ and the embedded $t$ are always identical, as the output of a text encoder (i.e., the candidate filtering model) shall retain the dimension of the input coming through the SA layers (see Figure 3). This characteristic makes merging the outputs of the two encoders within each transformer layer straightforward, which is discussed further below.

We use a default 12-layer transformer for each encoder. Within each transformer layer, both encoders cross-attend the inputs introduced above with the sequential feature of an arbitrary $I'_T$. The intuition is to allow $I'_T$ to separately attend to the two elements in each query $q$ for relevancy, namely $t$ and $I_R$. For

Encoder-$t$, the two entities entering the cross-attention (CA) layer are self-explanatory. For Encoder-$z_t$, however, we opt for using $z_t$ as a surrogate of $I_{\mathrm{R}}$. The main reason is the GPU memory limit, as it is intractable to perform image-image cross attention between $I_{\mathrm{R}}$ and $I'_{\mathrm{T}}$ with the default $L_{\boldsymbol{v}} = 577$ during training. Although spatial pooling can be used to reduce the length of the input $I_{\mathrm{R}}$ sequence, we empirically find it inferior, potentially due to the loss of information in pooling, which is discussed further in Section 4.3. As an alternative, $z_t$ can be viewed as an embedding that contains sufficient $I_{\mathrm{R}}$ information and is readily available, since it has been pre-computed in the previous stage from the query pair $q$. A bonus of using $z_t$ is that we can easily merge the cross-attended features, as it shares the same dimensionality as $t$ at all times. Empirically, we confirm that our design choices yield better results.

Figure 3 depicts the internal of the transformer layer of the re-ranking model. As illustrated, we merge the outputs of the CA layers from the two encoders in each transformer layer. Specifically, given the outputs of the encoders after the CA layers, the merging is performed as an average pooling in the first six layers, and a concatenation followed by a simple MLP in the last six layers. The merged feature is then passed into a residual connection, followed by the FF layers. Regarding weight-sharing across layers in each encoder, we opt for having separate weights of SA and CA layers within each encoder, while sharing the weights of FF layers to account for the different inputs passing through the SA and CA layers. We point out that due to the residual connections (Figure 3), the outputs of the two encoders after the final transformer block are different in values, even though the FF layers are of the same weights.

We formulate the re-ranking as a scoring task — among the set of candidate images score the true target higher than all other negative images. For each sequential output from either encoder, we extract the `[CLS]` token at the front as the summarized feature. We then concatenate the two `[CLS]` outputs from two encoders and use a two-layer MLP as the scorer head, which resembles the BLIP image-text matching (ITM) pre-training task (Li et al., 2022) setup.

### 3.3 Training Pipeline

### 3.3.1 Candidate Filtering

Our filtering model follows the contrastive learning pipeline (Radford et al., 2021) with a batch-based classification loss (Vo et al., 2019) commonly adopted in previous work (Liu et al., 2024; Levy et al., 2023). Specifically, in training, given a batch size of $B$, the features of the $i$-th query $\langle I_{\mathrm{R}}^i, t^i \rangle$ with its ground-truth target $I_{\mathrm{T}}^i$, we formulate the loss as:

$$\mathcal{L}_{\mathrm{Filtering}} = -\frac{1}{B} \sum_{i=1}^{B} \log \left[ \frac{\exp\left[ \lambda \cdot \kappa\left( f_\theta(I_{\mathrm{R}}^i, t^i), I_{\mathrm{T}}^i \right) \right]}{\sum_{j=1}^{B} \exp\left[ \lambda \cdot \kappa\left( f_\theta(I_{\mathrm{R}}^i, t^i), I_{\mathrm{T}}^j \right) \right]} \right], \tag{1}$$

where $f_\theta$ is the candidate filtering model parameterized by $\theta$, $\lambda$ is a learnable temperature parameter following (Radford et al., 2021), and $\kappa(\cdot, \cdot)$ is the similarity kernel as cosine similarity.

In inference, the model ranks all candidate images $I'_{\mathrm{T}}$ for each query via the same similarity kernel $\kappa(\cdot, \cdot)$. We then pick the top-$K$ list for each query for the second-stage re-ranking.

### 3.3.2 Candidate Re-ranking

The re-ranking model is trained with a similar contrastive loss as discussed above. Specifically, for each $\langle I_{\mathrm{R}}^i, t^i, I_{\mathrm{T}}^i \rangle$ triplet, we extract the predicted logit and contrast it against all other $\langle I_{\mathrm{R}}^i, t^i, I_{\mathrm{T}}^j \rangle$ with $i \neq j$, essentially creating $(B-1)$ negatives for each positive triplet. The loss is formulated as:

$$\mathcal{L}_{\mathrm{Re\text{-}ranking}} = -\frac{1}{B} \sum_{i=1}^{B} \log \left[ \frac{\exp\left[ f_\gamma(I_{\mathrm{R}}^i, t^i, I_{\mathrm{T}}^i) \right]}{\sum_{j=1}^{B} \exp\left[ f_\gamma(I_{\mathrm{R}}^i, t^i, I_{\mathrm{T}}^j) \right]} \right], \tag{2}$$

where $f_\gamma$ is the candidate re-ranking model parameterized by $\gamma$.

Note that in training, we randomly sample negatives within the same batch to form triplets. Therefore, the choice of $K$ does not affect the training process. We empirically find this yielding better performance than training only on the top-$K$ negatives, with the benefit of not relying on a filtered candidate list for training. Incidentally, it is also more efficient, as we do not need to independently load negatives for each query. During inference, the model only considers, for each query, the selected top-$K$ candidates and ranks them by the predicted logits.

## 4 Experiments

### 4.1 Experimental Setup

#### 4.1.1 Datasets

Following previous work, we consider two datasets in different domains. **Fashion-IQ** (Wu et al., 2021) is a dataset of fashion products in three categories, namely `Dress`, `Shirt`, and `Toptee`, which form over 30k triplets with 77k images. The annotations are collected from human annotators and are overall concise. **CIRR** (Liu et al., 2021) is proposed to specifically study the fine-grained visiolinguistic cues and implicit human agreements. It contains 36k pairs of queries with human-generated annotations, where images often contain rich object interactions (Suhr et al., 2019).[1]

#### 4.1.2 Evaluation Metrics

We follow previous work to report our results in Recall@$K$, that is the percentage of queries whose true target is ranked to be among the top-$K$ candidates. For Fashion-IQ, we assess the performance with Recall@10 and 50 on each category. Such choices of $K$ values account for the possible false negatives in the candidates. On CIRR, we report Recall@1, 5, 10, and 50. We additionally record Recall$_{\text{subset}}$@$K$, where the candidates are limited to a pre-defined set of five with high similarities. The set of five candidates contains no false negatives, making this metric more suitable to study fine-grained reasoning ability.

For Fashion-IQ, we report results on the validation split, as the ground truths of the test split remain nonpublic. For CIRR, we report our main results on the test split obtained from the evaluation server[2].

#### 4.1.3 Implementation Details

We adopt the standard image pre-processing and model configurations of BLIP encoders (Li et al., 2022). Except for image padding, which we follow Baldrati et al. (2022a) with a padding ratio of 1.25. Image resolution is set to $384 \times 384$. We initialize the image and text encoders with the `BLIP w/ ViT-B` pre-trained weights. In both stages, we freeze the ViT image encoder and only finetune the text encoders due to the GPU memory limits.

For all models in both stages, we use AdamW (Loshchilov & Hutter, 2019) with an initial learning rate of $2 \times 10^{-5}$, a weight decay of 0.05, and a cosine learning rate scheduler (Loshchilov & Hutter, 2017) with its minimum learning rate set to 0.

For candidate filtering (first stage) model, we train with a batch size of 512 for 10 epochs on both Fashion-IQ and CIRR. For candidate re-ranking (second stage) model, we reduce the batch size to 16 due to the GPU memory limit, as it requires exhaustively pairing up queries with each candidate. For Fashion-IQ, we train the re-ranking model for 50 epochs, for CIRR, we train it for 80 epochs.

All experiments are conducted on a single NVIDIA A100 80G with PyTorch while enabling automatic mixed precision (Micikevicius et al., 2018). We base our implementation on the BLIP codebase[3].

---

[1]Both datasets are publicly released under the MIT License, which allows distributions and academic usages.
[2]https://cirr.cecs.anu.edu.au/test_process/
[3]https://github.com/salesforce/BLIP

| | | Dress | | Shirt | | Toptee | | Average | | Avg. |
|---|---|---|---|---|---|---|---|---|---|---|
| | Methods | R@10 | R@50 | R@10 | R@50 | R@10 | R@50 | R@10 | R@50 | Metric |
| 1 | MRN (Kim et al., 2016) | 12.32 | 32.18 | 15.88 | 34.33 | 18.11 | 36.33 | 15.44 | 34.28 | 24.86 |
| 2 | FiLM (Perez et al., 2018) | 14.23 | 33.34 | 15.04 | 34.09 | 17.30 | 37.68 | 15.52 | 35.04 | 25.28 |
| 3 | TIRG (Vo et al., 2019) | 14.87 | 34.66 | 18.26 | 37.89 | 19.08 | 39.62 | 17.40 | 37.39 | 27.40 |
| 4 | Relationship (Santoro et al., 2017) | 15.44 | 38.08 | 18.33 | 38.63 | 21.10 | 44.77 | 18.29 | 40.49 | 29.39 |
| 5 | CIRPLANT (Liu et al., 2021) | 14.38 | 34.66 | 13.64 | 33.56 | 16.44 | 38.34 | 14.82 | 35.52 | 25.17 |
| 6 | CIRPLANT w/OSCAR (Liu et al., 2021) | 17.45 | 40.41 | 17.53 | 38.81 | 21.64 | 45.38 | 18.87 | 41.53 | 30.20 |
| 7 | VAL w/GloVe (Chen et al., 2020a) | 22.53 | 44.00 | 22.38 | 44.15 | 27.53 | 51.68 | 24.15 | 46.61 | 35.40 |
| 8 | CurlingNet (Yu et al., 2020) | 24.44 | 47.69 | 18.59 | 40.57 | 25.19 | 49.66 | 22.74 | 45.97 | 34.36 |
| 9 | DCNet (Kim et al., 2021) | 28.95 | 56.07 | 23.95 | 47.30 | 30.44 | 58.29 | 27.78 | 53.89 | 40.84 |
| 10 | CoSMo (Lee et al., 2021) | 25.64 | 50.30 | 24.90 | 49.18 | 29.21 | 57.46 | 26.58 | 52.31 | 39.45 |
| 11 | MAAF (Dodds et al., 2020) | 23.8 | 48.6 | 21.3 | 44.2 | 27.9 | 53.6 | 24.3 | 48.8 | 36.6 |
| 12 | ARTEMIS (Delmas et al., 2022) | 25.68 | 51.25 | 28.59 | 55.06 | 21.57 | 44.13 | 25.25 | 50.08 | 37.67 |
| 13 | SAC w/BERT (Jandial et al., 2022) | 26.52 | 51.01 | 28.02 | 51.86 | 32.70 | 61.23 | 29.08 | 54.70 | 41.89 |
| 14 | AMC (Zhu et al., 2023) | 31.73 | 59.25 | 30.67 | 59.08 | 36.21 | 66.06 | 32.87 | 61.64 | 47.25 |
| 15 | CLIP4CIR (Baldrati et al., 2022b) | 33.81 | 59.40 | 39.99 | 60.45 | 41.41 | 65.37 | 38.32 | 61.74 | 50.03 |
| 16 | BLIP4CIR (Liu et al., 2024) | 42.09 | 67.33 | 41.76 | 64.28 | 46.61 | 70.32 | 43.49 | 67.31 | 55.40 |
| 17 | FAME-ViL† (Han et al., 2023) | 42.19 | 67.38 | 47.64 | 68.79 | 50.69 | 73.07 | 46.84 | 69.75 | 58.29 |
| 18 | CASE (Levy et al., 2023) | 47.77 | 69.36 | 48.48 | 70.23 | 50.18 | 72.24 | 48.79 | 70.68 | 59.74 |
| 19 | Ours **F** | 43.78 | 67.38 | 45.04 | 67.47 | 49.62 | 72.62 | 46.15 | 69.15 | 57.65 |
| 20 | Ours $\mathbf{R}_{100}$ | **48.14** | **71.34** | **50.15** | **71.25** | **55.23** | **76.80** | **51.17** | **73.13** | **62.15** |

Table 1: Fashion-IQ, validation split. We report Average Metric (Recall$_{Avg}$@10+Recall$_{Avg}$@50)/2 as in (Wu et al., 2021). Rows 1-2 are cited from (Wu et al., 2021). †: Methods trained with additional data in a multi-task setup. **F** (shaded) denotes candidate filtering model, $\mathbf{R}_K$ denotes candidate re-ranking model with results obtained on the top-$K$ filtered results from **F**. For Fashion-IQ we use top-100, which has a ground truth coverage of 77.24%, 75.86% and 81.18% for `Dress`, `Shirt` and `Toptee` categories respectively. Best numbers (resp. second-best) are in **black** (resp. underlined).

| | | Recall@$K$ | | | | Recall$_{Subset}$@$K$ | | | Avg. |
|---|---|---|---|---|---|---|---|---|---|
| | Methods | $K=1$ | $K=5$ | $K=10$ | $K=50$ | $K=1$ | $K=2$ | $K=3$ | Metric |
| 1 | TIRG (Vo et al., 2019) | 14.61 | 48.37 | 64.08 | 90.03 | 22.67 | 44.97 | 65.14 | 35.52 |
| 2 | TIRG+LastConv (Vo et al., 2019) | 11.04 | 35.68 | 51.27 | 83.29 | 23.82 | 45.65 | 64.55 | 29.75 |
| 3 | MAAF (Dodds et al., 2020) | 10.31 | 33.03 | 48.30 | 80.06 | 21.05 | 41.81 | 61.60 | 27.04 |
| 4 | MAAF+BERT (Dodds et al., 2020) | 10.12 | 33.10 | 48.01 | 80.57 | 22.04 | 42.41 | 62.14 | 27.57 |
| 5 | MAAF−IT (Dodds et al., 2020) | 9.90 | 32.86 | 48.83 | 80.27 | 21.17 | 42.04 | 60.91 | 27.02 |
| 6 | MAAF−RP (Dodds et al., 2020) | 10.22 | 33.32 | 48.68 | 81.84 | 21.41 | 42.17 | 61.60 | 27.37 |
| 7 | CIRPLANT (Liu et al., 2021) | 15.18 | 43.36 | 60.48 | 87.64 | 33.81 | 56.99 | 75.40 | 38.59 |
| 8 | CIRPLANT w/OSCAR (Liu et al., 2021) | 19.55 | 52.55 | 68.39 | 92.38 | 39.20 | 63.03 | 79.49 | 45.88 |
| 9 | ARTEMIS (Delmas et al., 2022) | 16.96 | 46.10 | 61.31 | 87.73 | 39.99 | 62.20 | 75.67 | 43.05 |
| 10 | CLIP4CIR (Baldrati et al., 2022b) | 38.53 | 69.98 | 81.86 | 95.93 | 68.19 | 85.64 | 94.17 | 69.09 |
| 11 | BLIP4CIR (Liu et al., 2024) | 40.15 | 73.08 | 83.88 | 96.27 | 72.10 | 88.27 | 95.93 | 72.59 |
| 12 | CASE (Levy et al., 2023) | 48.00 | 79.11 | 87.25 | **97.57** | 75.88 | 90.58 | 96.00 | 77.50 |
| 13 | CASE Pre-LaSCo.Ca.† (Levy et al., 2023) | 49.35 | 80.02 | 88.75 | 97.47 | 76.48 | 90.37 | 95.71 | 78.25 |
| 14 | Ours **F** | 44.70 | 76.59 | 86.43 | 97.18 | 75.02 | 89.92 | 95.64 | 75.81 |
| 15 | Ours $\mathbf{R}_{50}$ | **50.55** | **81.75** | **89.78** | 97.18 | **80.04** | **91.90** | **96.58** | **80.90** |

Table 2: CIRR, test split. We pick our best-performing model on the validation split and submit results online for testing. We report the Average Metric (Recall@5+Recall$_{Subset}$@1)/2 as in (Liu et al., 2021). Rows 1-8 are cited from (Liu et al., 2021). †: Methods trained with additional pre-training data. **F** (shaded) denotes candidate filtering model, $\mathbf{R}_K$ denotes candidate re-ranking model with results obtained on the top-$K$ filtered results from **F**. For CIRR we use top-50, which has a ground truth coverage of 97.18%. Best numbers (resp. second-best) are in **black** (resp. underlined).

## 4.2 Performance Comparison with State-of-the-art

### 4.2.1 Results on Fashion-IQ

Table 1 compares the performance on Fashion-IQ. We note that our re-ranking model (row 20) outperforms all existing methods consistently across three categories. Impressively, the performance increase is notable when compared to CASE (row 18), a method that also uses BLIP encoders. This suggests that our two-stage design, particularly the explicit query-specific candidate re-ranking, is beneficial to the task.

| | Methods | Dress | | Shirt | | Toptee | | Average | | Avg. |
|---|---|---|---|---|---|---|---|---|---|---|
| | | R@10 | R@50 | R@10 | R@50 | R@10 | R@50 | R@10 | R@50 | Metric |
| | Ours $\mathbf{R}_{100}$ | **48.14** | **71.34** | **50.15** | **71.25** | **55.23** | **76.80** | **51.17** | **73.13** | **62.15** |
| 1 | *w/o.* Encoder-$t$ | 41.75 | 68.17 | 42.64 | 68.11 | 47.83 | 74.35 | 44.07 | 70.21 | 57.14 |
| 2 | *w/o.* Encoder-$z_t$ | 37.48 | 66.83 | 39.16 | 65.75 | 46.25 | 72.62 | 40.96 | 68.40 | 54.68 |
| 3 | *w.* $\text{Ref}_{\text{CLS}}$ | 46.55 | 71.24 | 47.84 | 70.07 | 54.36 | 75.83 | 49.59 | 72.38 | 60.98 |
| 4 | *w.* $\text{Ref}_{\text{CLS + Spatial-6}\times 6}$ | 48.04 | 71.10 | 48.04 | 70.31 | 54.82 | 76.39 | 50.30 | 72.60 | 61.45 |
| 5 | Full-MLP merge | 47.00 | 70.80 | 47.79 | 69.63 | 54.61 | 76.54 | 49.80 | 72.32 | 61.06 |
| 6 | Dual Feed-forward | 44.67 | 69.71 | 46.37 | 69.97 | 52.83 | 76.39 | 46.96 | 72.02 | 59.99 |

Table 3: Ablation studies on Fashion-IQ, validation split. We report Average Metric as $(\text{Recall}_{\text{Avg}}@10+\text{Recall}_{\text{Avg}}@50)/2$. Shaded is our candidate re-ranking model as in Table 1 row 20. Rows 1–2 ablate the utility of the dual-encoder design. Rows 3–4 examine the difference between using $z_t$ and the reference image features in Encoder-$z_t$ in Figure 2 (right). In row 4, we choose $\text{Ref}_{\text{CLS + Spatial-6}\times 6}$ to showcase the performance under minimum information loss with pooling, as the hardware cannot accommodate a longer sequence of image input. Rows 5–6 test architectural designs. Best results are in **black**.

| | Methods | Recall@$K$ | | | | Recall$_{\text{Subset}}$@$K$ | | | Avg. |
|---|---|---|---|---|---|---|---|---|---|
| | | $K=1$ | $K=5$ | $K=10$ | $K=50$ | $K=1$ | $K=2$ | $K=3$ | Metric |
| | Ours $\mathbf{R}_{50}$ | 53.24 | 83.11 | 90.03 | **97.08** | 80.44 | 92.54 | 97.01 | **81.78** |
| 1 | *w/o.* Encoder-$t$ | 46.21 | 79.57 | 89.33 | **97.08** | 75.25 | 89.72 | 95.72 | 77.41 |
| 2 | *w/o.* Encoder-$z_t$ | 43.51 | 75.25 | 84.24 | **97.08** | 80.34 | 91.68 | 96.72 | 77.79 |
| 3 | *w.* $\text{Ref}_{\text{CLS + Spatial-6}\times 6}$ | 52.98 | **83.23** | **90.48** | **97.08** | 79.55 | 90.06 | 96.70 | 81.39 |
| 4 | Full-MLP merge | 52.91 | 82.64 | 90.15 | **97.08** | 79.60 | 92.25 | 96.89 | 81.12 |
| 5 | Dual Feed-forward | **54.20** | 82.80 | 90.39 | **97.08** | 80.22 | 91.89 | 96.53 | 81.51 |

Table 4: Ablation studies on CIRR, validation split. We report the Average Metric $(\text{Recall}@5+\text{Recall}_{\text{Subset}}@1)/2$. Experiments conducted following Table 3 on Fashion-IQ. Shaded is our candidate re-ranking model as in Table 2 row 15. Note the values differ for the test and validation splits. Best results are in **black**.

Regarding our first stage filtering model (row 19), we achieve a performance slightly behind CASE. As discussed in Section 3, we share a similar, default BLIP-based (Li et al., 2022) architecture and training pipeline as CASE. Upon examining the ablation studies by Levy et al. (2023), we conjecture that the lower performance is mainly because we adopt a different loss and do not finetune the ViT image encoder alongside due to hardware limits[4]. We note that our first stage model is primarily intended to obtain the top-$K$ filtered candidate list, and that the main focus of our work — the re-ranking model — outperforms all existing methods by a large margin.

### 4.2.2 Results on CIRR

Table 2 compares the performance on CIRR. Overall, we observe a similar trend in performance increase as in Fashion-IQ. This includes the performance comparison between our filtering model (row 14) and CASE (Levy et al., 2023) (row 12), as discussed above. We notice that our re-ranked results (row 15) outperform all previous methods, including models that are based on BLIP and pre-trained on additional data of large scales (row 13). This demonstrates that our design more effectively harnesses the information within the input entities than existing work through explicit query-candidate reasoning.

---

[4]We are unable to reproduce or assess their results at the time of writing as the code for CASE is not publicly released.

| | | Dress | | Shirt | | Toptee | | Average | | Avg. |
|---|---|---|---|---|---|---|---|---|---|---|
| | Methods | R@10 | R@50 | R@10 | R@50 | R@10 | R@50 | R@10 | R@50 | Metric |
| 1 | **F** (candidate filtering) | 43.78 | 67.38 | 45.04 | 67.47 | 49.62 | 72.62 | 46.15 | 69.15 | 57.65 |
| 2 | $\mathbf{R}_{50}$ | 48.24 | 67.38 | 50.15 | 67.47 | 54.56 | 72.62 | 50.98 | 69.15 | 60.07 |
| 3 | $\mathbf{R}_{70}$ | 48.14 | 69.66 | 50.59 | 69.09 | 54.87 | 75.83 | 51.20 | 71.52 | 61.36 |
| 4 | $\mathbf{R}_{90}$ | 48.14 | 70.90 | 50.15 | 70.76 | 55.07 | 76.13 | 51.12 | 72.60 | 61.86 |
| 5 | $\mathbf{R}_{100}$ | 48.14 | 71.34 | 50.15 | 71.25 | 55.23 | 76.80 | 51.17 | 73.13 | 62.15 |
| 6 | $\mathbf{R}_{150}$ | 48.09 | 71.49 | 50.20 | 71.49 | 55.28 | 77.41 | 51.19 | 73.46 | 62.33 |
| 7 | $\mathbf{R}_{200}$ | 47.89 | 71.44 | 50.15 | 71.00 | 55.38 | 77.41 | 51.14 | 73.28 | 62.21 |

Table 5: Fashion-IQ, validation split. $\mathbf{R}_K$: Ablation on $K$ value in candidate re-ranking. **F** (row 1): the candidate filtering model (attached as a reference to compare against), as in Table 1 row 19. Shaded: our choice of $K$ when reporting the main results in Table 1 row 20.

| | | Recall@$K$ | | | | Recall$_{\text{Subset}}$@$K$ | | | Avg. |
|---|---|---|---|---|---|---|---|---|---|
| | Methods | $K=1$ | $K=5$ | $K=10$ | $K=50$ | $K=1$ | $K=2$ | $K=3$ | Metric |
| 1 | **F** (candidate filtering) | 46.83 | 78.59 | 88.04 | 97.08 | 76.11 | 90.65 | 96.05 | 77.53 |
| 2 | $\mathbf{R}_{30}$ | 53.03 | 83.16 | 90.62 | 95.26 | 80.44 | 92.54 | 97.01 | 81.80 |
| 3 | $\mathbf{R}_{40}$ | 52.98 | 82.83 | 90.29 | 96.27 | 80.44 | 92.54 | 97.01 | 81.64 |
| 4 | $\mathbf{R}_{50}$ | 53.24 | 83.11 | 90.03 | 97.08 | 80.44 | 92.54 | 97.01 | 81.78 |
| 5 | $\mathbf{R}_{100}$ | 52.88 | 82.92 | 90.07 | 97.90 | 80.44 | 92.54 | 97.01 | 81.68 |
| 6 | $\mathbf{R}_{150}$ | 52.91 | 82.85 | 90.05 | 98.04 | 80.44 | 92.54 | 97.01 | 81.65 |

Table 6: CIRR, validation split. $\mathbf{R}_K$: Ablation on $K$ value in candidate re-ranking. **F** (row 1): the candidate filtering model (attached as a reference to compare against), as in Table 2 row 14. Shaded: our choice of $K$ when reporting the main results in Table 2 row 15. Note the values differ for the test and validation splits.

## 4.3 Ablation Studies

### 4.3.1 Key Design Choices

In Table 3, we test several variants of the re-ranking model to verify our design choices. We report performance on the Fashion-IQ validation split for all experiments.

We begin with assessing the necessity of our dual-encoder setup, as shown in Table 3 rows 1 and 2. Given that $z_t$ is obtained from both the text $t$ and the reference image $I_{\text{R}}$ (Figure 2 left), one might question if explicit text information, i.e., Encoder-$t$, is still needed in the re-ranking step. To this end, in row 1, we demonstrate that the performance would decrease significantly if Encoder-$t$ is removed from the setup. Subsequently, in row 2, we validate that Encoder-$z_t$ is also crucial to the performance, which suggests that the reference image information embedded in $z_t$ is meaningful for the prediction. However, this alone does not fully justify our scheme of using $z_t$ as a surrogate of $I_{\text{R}}$, thus allowing interactions between the reference and candidate images. Recall what was discussed in Section 3, our motivation for adopting $z_t$ instead of the embedding of $I_{\text{R}}$ is that GPU memory consumption prohibits direct image-image cross-attention, unless certain spatial pooling is applied to the reference image. To validate that using $z_t$ is a better alternative to the pooling methods, in rows 3 and 4, we show that the performance decreases when replacing $z_t$ with such pooled features.

We additionally show two variants related to our design choices. Row 5 replaces the first six merging layers from the average pooling to MLP, while row 6 removes the weight-sharing of the FF layers. We note a consistent performance decrease in both cases.

Table 4 shows the ablation studies conducted on CIRR, which complements the same set of experiments performed on Fashion-IQ. We find that both the text and reference image play an important role in CIRR (rows 1 and 2) as in Fashion-IQ, and our choice of using $z_t$ as a surrogate for it is validated (row 3). Regarding the ablated design choices of the architecture (rows 4 and 5), we notice they bear a slightly smaller impact on CIRR than on Fashion-IQ, but our design still yields better overall performance in the Recall$_{\text{Subset}}$ and Average Metric.

Interestingly, when combining the results shown in rows 1 and 2, we discover that Encoder-$t$, i.e., explicit text information, is more important to the Recall$_{\text{Subset}}$ metric, while Encoder-$z_t$, along with the embedded reference image, is more crucial to the global Recall metric. This is thought to be caused by the fact that Recall$_{\text{Subset}}$ only considers a selected group of five candidates of high similarities, hence the information within the reference image contributes less to the retrieval. Meanwhile, the need of using visual cues in the reference image to exclude negatives remains vital to the global Recall as the candidates are not pre-selected per their visual similarities in this scenario. Indeed, previous (Liu et al., 2021) as well as concurrent work (Levy et al., 2023; Vaze et al., 2023) observe a similar case in their ablation studies.

### 4.3.2 K values in Candidate Re-ranking

Tables 5 and 6 study the effect of varying $K$ on the re-ranking model. Recall that $K$ is the number of samples taken from the first stage to be re-ranked. As discussed in Section 3.3, $K$ only affects inference but not training.

Given that increasing $K$ effectively increases the ground truth coverage (see definition in Section 3.1 and detailed statistics in Section A.2), one could reasonably expect that a larger $K$ yields higher performance. However, we note that a higher $K$ would also lead to more negative candidates, which potentially impact the performance should the re-ranking model fails to properly rank them. To this end, a trade-off exists between the two contributing factors.

For Fashion-IQ, we see a general trend of performance increase while increasing $K$, except for when $K = 200$ (Table 5 row 7). For CIRR (Table 6), we observe minor performance differences when varying $K$, which is unsurprising given the high ground truth coverage in general (see Table 2 caption). Consequently, we could only notice a clear trend in Recall@50. Here, we note that $K$ does not affect validating on Recall$_{\text{Subset}}$, as such a metric only concerns five pre-determined candidates per query.

As discussed in Section 3.1, we have not handpicked $K$ per training instance. Instead, for every dataset, we globally select $K$ based on the empirical balance between the ground truth coverage and inference cost.

### 4.4 Inference Time

One obvious limitation of our method is the inference time of the re-ranking model, as it requires exhaustive pairing of the query and top-$K$ candidates. Several factors contribute to the case, including the size of the validation/test split of the dataset, choice of $K$, as well as the general length of the input text, which affects the efficiency of the attention layers within the transformer. We observe that compared to a traditional vector distancing-based method, in this case, our filtering model, the inference time of the re-ranking step is increased by approximately eight times on Fashion-IQ and 35 times on CIRR. Qualitatively, with our default choices of $K$ in Tables 1 and 2, it takes around 0.10 seconds to infer on a query for Fashion-IQ and 0.11 seconds for CIRR, which amounts to approximately 10 and 7.5 minutes for the validation split of the two datasets respectively. See Section A.3 for a detailed analysis on inference time *vs.* the choice of $K$.

We note that our focus of this work is on achieving higher performance through model architectural design and better use of input information, and is not optimized for applications where processing speed is more important than retrieval accuracy. This is in line with most existing work on vision-and-language, such as BLIP (Li et al., 2022), where the task of image-text retrieval is also studied in a non real-time fashion. We point out that the additional cost results in a significant increase in performance — around 5% absolute in average Recall compared to the pre-filtered results (Tables 1 and 2, between the last two rows).

### 4.5 Qualitative Results

We present several retrieved results on CIRR in Figure 4, where we show the pipeline of filtering followed by re-ranking. For **(a)** and **(b)**, we note that explicit text-candidate pairing can be more beneficial in cases where new elements are added (i.e., "trees", "two people" and "cat"), as the re-ranking model can readily identify concepts within each candidate, and assess its correlations between the text. We specifically point to **(b)**, where with initial filtering, only one candidate among the top-6 contains a cat as the text describes. After re-ranking, three candidates with cat are brought forward, with the true target ranked the first. In

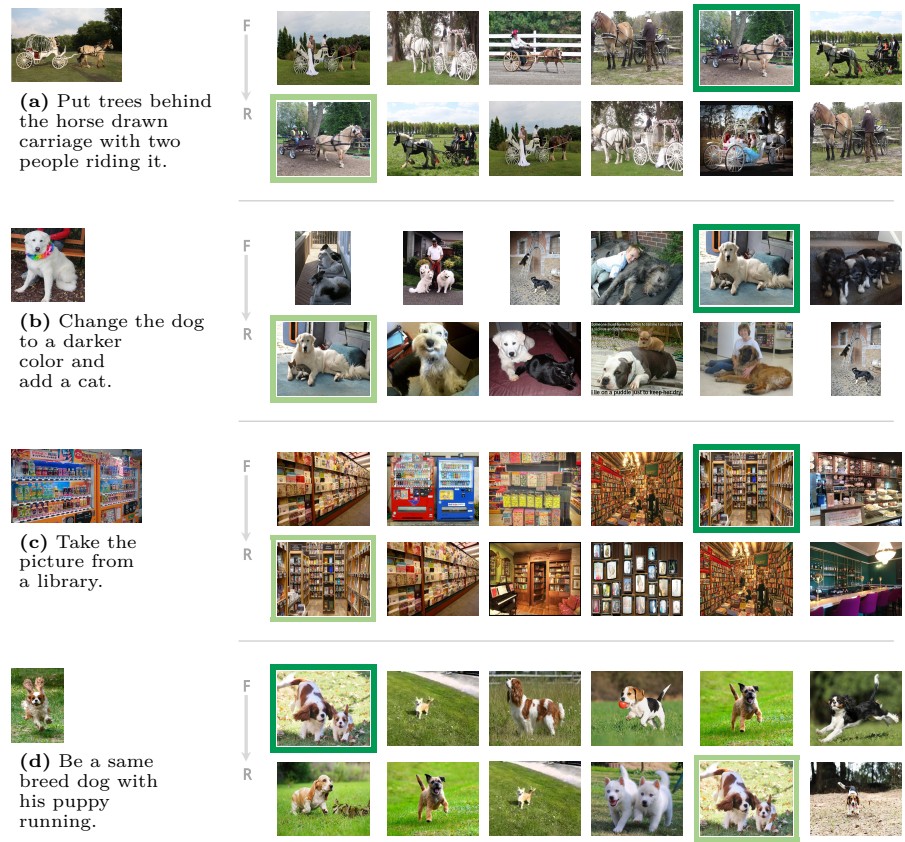

Figure 4: Qualitative examples on CIRR. For each sample, we showcase the query (left) with the filtered top-6 candidates (row **F**), followed by the re-ranked top-6 results (row **R**). True targets are in green frames. We demonstrate three cases where re-ranking brings the true target forward **(a-c)**, and one failure case **(d)**.

**(c)**, we show that our re-ranking model is also effective at recognizing global visual concepts such as scenes (i.e., library). Finally, we list a failure case of the re-ranking in **(d)**, where we observe that our re-ranking model fails to associate the concept of "with" with having two entities simultaneously in the image. As a result, the top-3 re-ranked candidates each only pictures a single puppy running. We additionally point out that the filtering module is effective at removing easy negatives. As shown in Figure 4 (row **F**) on each sample, the top-ranked candidates are already picturing similar objects or scenes as the reference. Qualitative examples on Fashion-IQ are demonstrated in Appendix A.1. A detailed quantitative analysis on the effect of re-ranking is presented in Section A.4, which complements discussions in this section.

## 5 Conclusion

We propose a two-stage method for composed image retrieval, which trades off the inference cost with a model that exhaustively pairs up queries with each candidate image. Our filtering module follows prior work and uses vector distances to quickly generate a small candidate set. We then design a dual-encoder architecture to assess the remaining candidates against the query for their relevancy. Both stages of our method are designed based on the existing vision-and-language pre-trained model. We experimentally show that our approach consistently outperforms existing methods on two popular benchmarks, Fashion-IQ and CIRR.

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

# A   Appendix

## A.1   Qualitative Examples on Fashion-IQ

Here, we provide qualitative examples on Fashion-IQ in Figure 5, which tells a similar story as in CIRR. We show cases where re-ranking brings the ground truth forward in **(a-d)**, along with failure cases in **(e-f)**. Note the sometimes ambiguous and abstract human annotations when describing fashion style changes (e.g., "more patriotic" in **a**), whose ground truth target can be difficult to estimate purely based on the query. In these conditions, explicitly reasoning over candidate images is particularly beneficial.

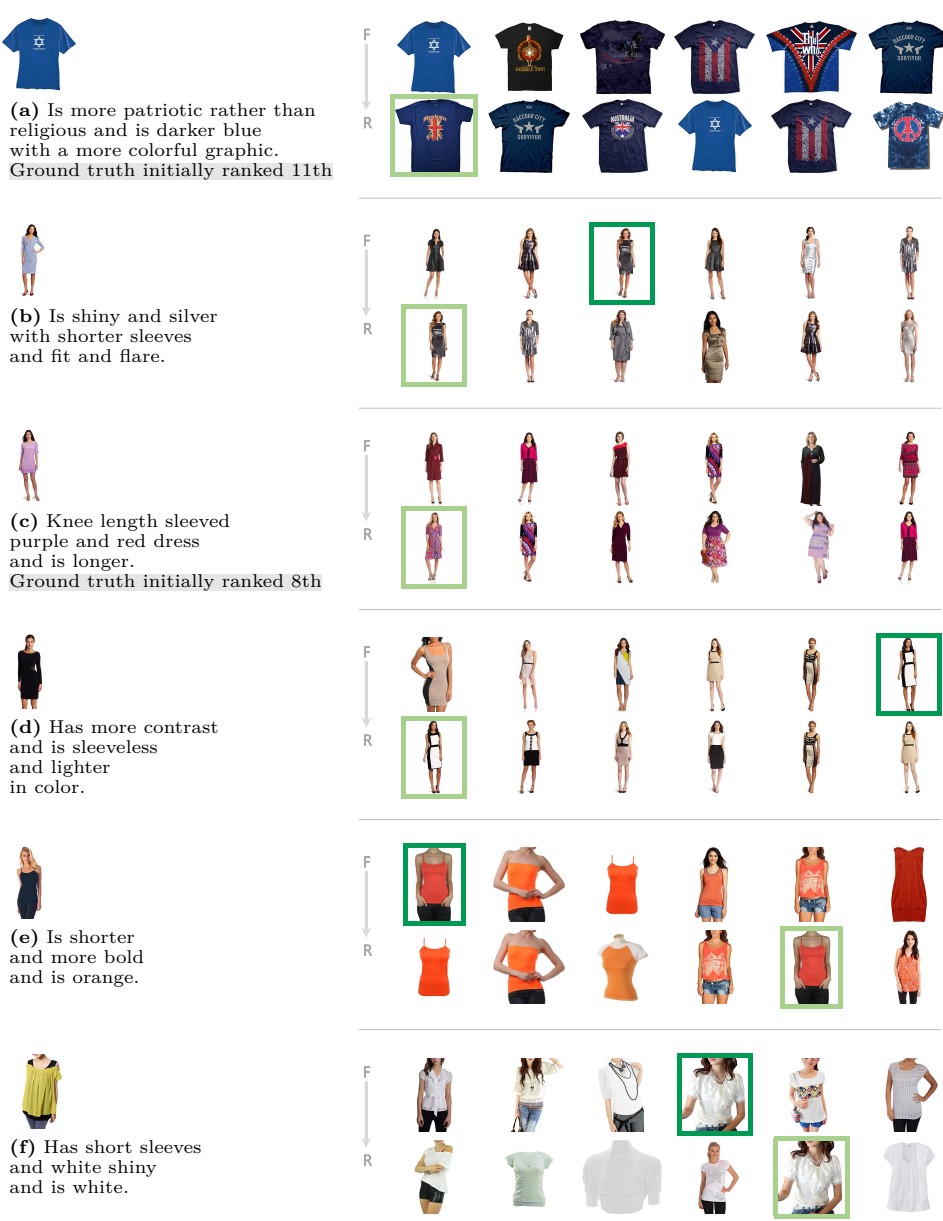

Figure 5: Qualitative examples on Fashion-IQ. For each sample, we showcase the query (left) with the filtered top-6 candidates (row **F**), followed by the re-ranked top-6 results (row **R**). Each query comes with two sentences of annotations which is joined by "and". True targets are in green frames. For examples with ground truth initially ranked beyond the top-6, we report their rankings below the annotation, as in **(a)** and **(c)**.

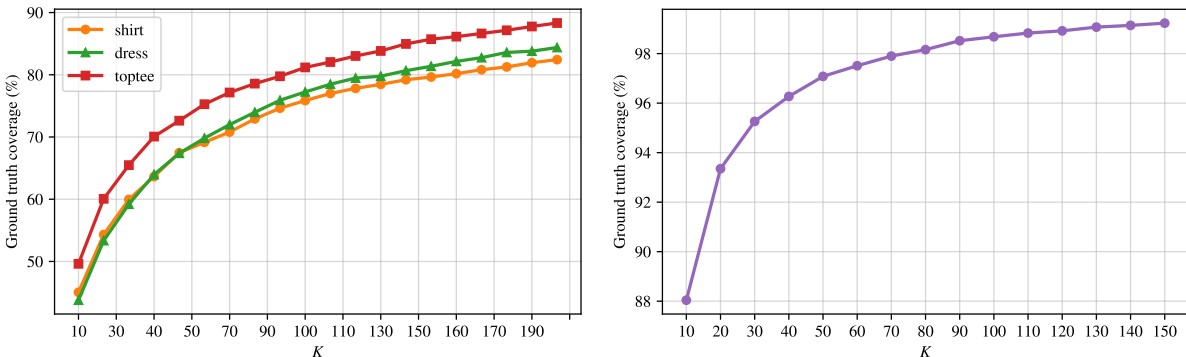

Figure 6: Ground truth coverage (in percentage) *vs.* $K$ for candidate re-ranking. **Left:** Fashion-IQ validation split with statistics of the three categories. **Right:** CIRR validation split. The ranges of $K$ follow Tables 5 and 6.

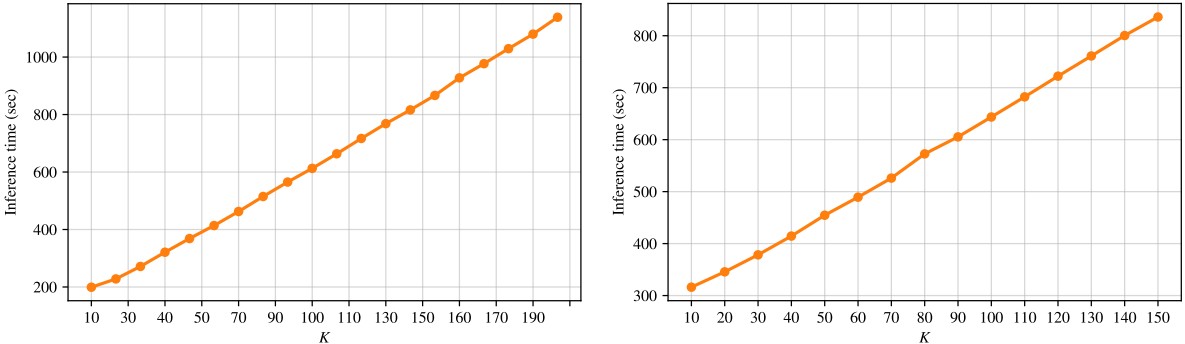

Figure 7: Inference time (in seconds) *vs.* $K$ for candidate re-ranking. **Left:** Fashion-IQ validation split. **Right:** CIRR validation split. The ranges of $K$ follow Tables 5 and 6.

## A.2 Analysis on Ground Truth Coverage *vs.* K

In Figure 6, we illustrate the relationship between the ground truth coverage and the choice of $K$, which complements discussions in Section 4.3.2.

## A.3 Analysis on Inference Time *vs.* K

In Figure 7, we study the relationship between the choice of $K$ and the inference time, which complements discussions in Section 4.4. We anticipate their relationship to be roughly linear. Here, we point out that regardless of the choice of $K$, there exist certain fixed overheads (such as model checkpoint loading). However, they only account for a small percentage of the total time. Therefore, the overall trend remains linear, as demonstrated in the figure.

## A.4 Quantitative Analysis on Candidate Re-ranking

Here, we perform a quantitative analysis of the effect of the re-ranking model. In Table 7, we report the average rankings of the targets in the validation split before and after the re-ranking (termed the initial and re-ranked rankings), along with their differences for both datasets. Globally, we observe that the targets are more highly ranked on average after re-ranking, which corresponds to the higher Recall@$K$ performance in Tables 1 and 2. We notice that on Fashion-IQ, the targets are on average brought forward by 6 to 7 in the indexes depending on the category, which is a significant improvement. Meanwhile, on CIRR, the absolute

|  | Initial ranking | Re-ranked ranking | Difference ($\Delta$) | Percentage decrease (%) |
|---|---|---|---|---|
| **Fashion-IQ** Dress | 28.74 | 22.25 | 6.49 | 22.58 |
| **Fashion-IQ** Shirt | 26.96 | 20.42 | 6.54 | 24.26 |
| **Fashion-IQ** Toptee | 25.81 | 18.74 | 7.07 | 27.39 |
| **CIRR** | 5.32 | 4.09 | 1.22 | 23.12 |

Table 7: The average initial and re-ranked rankings for targets in the validation splits of Fashion-IQ and CIRR. For Fashion-IQ, we report detailed statistics of the three categories. **Initial ranking**: rankings after the first stage filtering model. **Re-ranked ranking**: rankings after the second stage re-ranking model. **Difference** ($\Delta$): the difference between the two stages, where a positive value suggests that the targets are brought forward by the re-ranking model, which is favoured. **Percentage decrease**: calculated as (difference / initial ranking) $\times 100\%$. We consider the top-200 candidates per query for this analysis.

value is smaller. However, we note that the initial rankings of targets in CIRR are also much higher, leaving less room for improvement compared to Fashion-IQ. Overall, the percentage decrease in the target indexes on both datasets is over 20%, which validates that our re-ranking model is extremely effective.

Figure 8 further investigates the effectiveness of our re-ranking model, in particular, we wish to answer the question "How much benefits does the re-ranking stage bring to queries of different levels of difficulty?". To this end, we examine, for each dataset (and category in Fashion-IQ), the average difference ($\Delta$) in target indexes before and after re-rank *vs.* their initial rankings. We first observe that most of the $\Delta$ values are positive, suggesting that the targets are brought forward through re-ranking. This corroborates with the discussions above for Table 7. To take the analysis to a finer level, we point to cases where the initial rankings of the targets are extremely low (i.e., their indexes are high, which puts them on the right-hand side of the histograms). They are presumed hard cases, as the first stage filtering model failed at ranking the targets high. We note that our re-ranking model can often significantly improve the target rankings by over 50 or even 100 in indexes under such circumstances, as demonstrated by the high $\Delta$ values. This further confirms the strength of re-ranking. Interestingly, we also point out that the standard Recall@$K$ metric may not fully capture the benefit of the re-ranking in such cases, mostly because a target that is initially ranked extremely low may not be brought forward sufficiently enough, even if its index is improved by 50 or 100. This is especially true for CIRR where the metric considers Recall@5. Nevertheless, the improvements in such challenging cases still reveal the effectiveness of our re-ranking model.

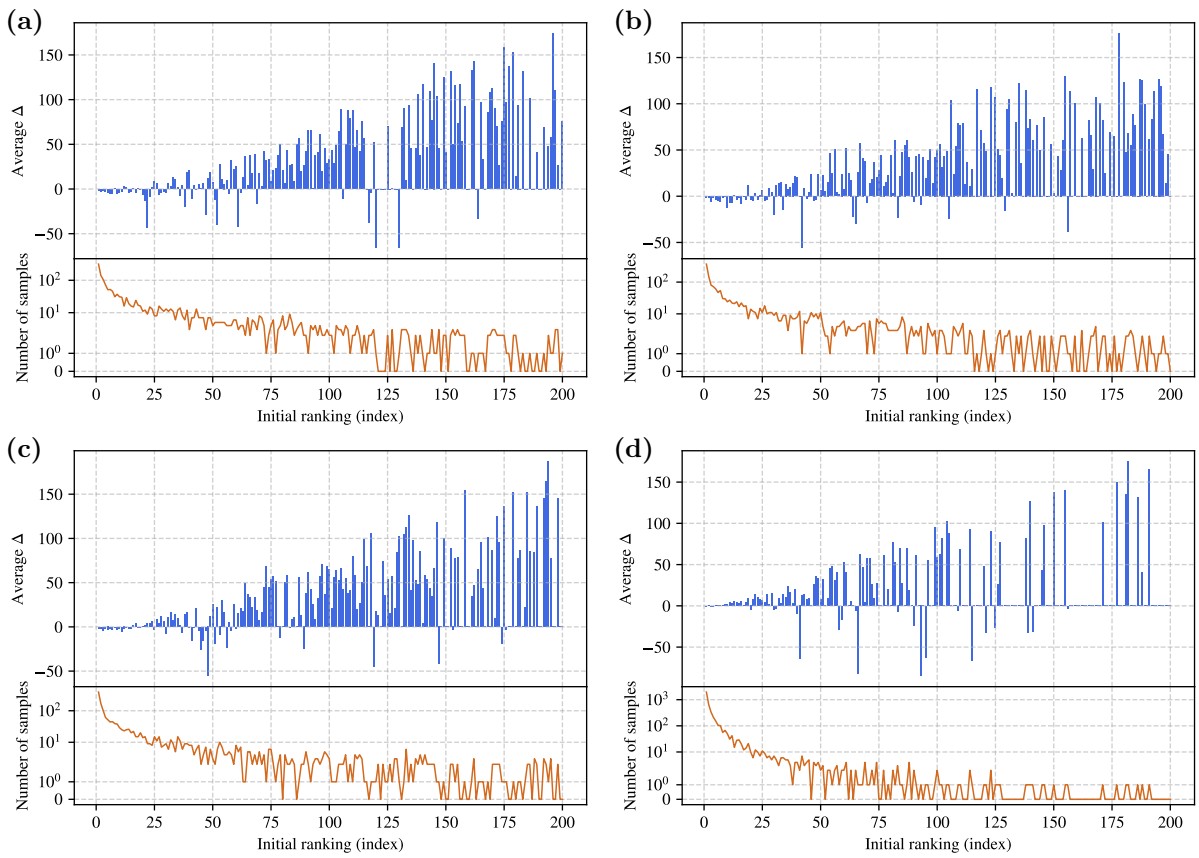

Figure 8: The initial rankings of targets *vs.* the difference in average ranking (noted as *Average* Δ) before and after the re-ranking stage. A positive *Average* Δ value suggests that the targets are brought forward by the re-ranking model, which is favoured. Likewise for a negative value. In each plot, we also show the number of queries counted for every initial ranking. **(a)** Fashion-IQ `Dress`. **(b)** Fashion-IQ `Shirt`. **(c)** Fashion-IQ `Toptee`. **(d)** CIRR. Analysis performed on the validation splits. We consider the top-200 candidates per query for this analysis, as in Table 7.

