# OpenReview forum: "Candidate Set Re-ranking for Composed Image Retrieval with Dual Multi-modal Encoder"
_TMLR — Accepted by TMLR_

### Review · Reviewer_HV9g · 2023-10-02

**Summary Of Contributions:**

In this research paper, the authors investigate the challenge of composed image retrieval. Traditional approaches involve computing multi-modal embeddings for both query and target images, subsequently returning candidates with the smallest cosine distance to the query. However, the authors argue that augmenting this process with a second re-ranking stage can significantly enhance performance.

The key insight lies in reformulating the re-ranking task as a scoring problem, wherein true-positive candidates are assigned higher scores, while other candidates receive lower scores. Extensive evaluations conducted on Fashion-IQ and CIRR datasets demonstrate that the proposed method yields substantial performance improvements over single-stage baselines. It is worth noting that this enhanced performance comes at the expense of relatively higher computational runtime, ranging from 8 to 35x.

**Audience:**

Yes

**Claims And Evidence:**

Yes

**Requested Changes:**

It would be great if the authors could respond to my comments on "Weaknesses".

**Strengths And Weaknesses:**

**Strengths:**

**Motivation:** The paper's motivation is commendable. Designing a reliable metric for single-stage retrieval methods that can simultaneously perform coarse-grained filtering and fine-grained ranking is a challenging task. Thus, the addition of a specialized re-ranking stage is a natural and beneficial choice.

**Clarity:** The paper is well-written and effectively communicates its key points. Figures 2 and 3 are of high quality and successfully convey essential information.

**Performance Improvement:** The proposed method demonstrates a significant performance improvement. For instance, in Table 1, it outperforms the single-stage baseline (Ours F) by approximately 4.5%. Additionally, it surpasses all other publications listed in the table.

**Ablation Analysis:** The authors have invested substantial effort in conducting ablation experiments to analyze their design choices thoroughly.

**Weaknesses:**

**Input Limitation:** It appears that the method requires both an input image and a text description, limiting its real-world applicability. It would be ideal if the proposed method could handle scenarios where only text inputs are provided, enhancing its versatility.

**Runtime Overhead:** The second stage introduces a substantial runtime overhead, resulting in a 5% performance gain. This trade-off may not be favorable, considering the speed overhead of 8-35x. Given this large runtime overhead, simpler alternatives such as ensembling might outperform the baseline significantly. Moreover, the proposed method's scalability to real-world datasets and applications may be hindered by this runtime overhead.

**Component Originality:** While the paper is easy to understand and well-written, it appears that most of its components are derived from existing methods. For example, the authors acknowledge that the objective is similar to and inspired by BLIP's ITM loss, raising questions about the novelty of the approach.

---

> ### Author Response · Authors · 2023-12-12
> **Response to review by reviewer HV9g**
>
> We thank reviewer HV9g for their comments. Please see below for our response.
>
> > Input Limitation: It appears that the method requires both an input image and a text description, limiting its real-world applicability. It would be ideal if the proposed method could handle scenarios where only text inputs are provided, enhancing its versatility.
>
> We follow previous work and adhere to the setup of the composed image retrieval task, which leverages an input of both image and text.
> We note that this task is worth studying, as it provides the users with more detailed controls over the retrieved target compared to conventional text-image retrieval.
>
> We acknowledge that other forms of image retrieval exist, and a method that can handle various application scenarios is practically appealing.
> However, our main focus and contribution is on the specific task of composed image retrieval.
>
> > Runtime Overhead: The second stage introduces a substantial runtime overhead, resulting in a 5% performance gain. This trade-off may not be favorable, considering the speed overhead of 8-35x. ... Moreover, the proposed method's scalability to real-world datasets and applications may be hindered by this runtime overhead.
>
> We acknowledge the inference cost of our method, as discussed in Section 4.4.
>
> We wish to point out that our method establishes a new benchmark for the task, though, it is not optimized for application scenarios where inference time is more important than retrieval accuracy. As discussed in Section 4.4, this is in line with most existing work on vision-and-language, where the task of text-image retrieval is also studied in a non-real-time fashion.
>
> We note that real-life applications where data is processed offline do exist, in which case, users can afford the inference cost and will likely appreciate a significant improvement in retrieval accuracy.
>
> For instance, our method can be used to clean up the candidate image corpus (removing irrelevant candidates) for researchers who wish to construct future datasets on this task.
>
> Ultimately, the user shall determine if the trade-off between the inference cost and the retrieval accuracy is worth it.
>
> > (continuing the above comment) Given this large runtime overhead, simpler alternatives such as ensemble might outperform the baseline significantly.
> >
> On the topic of ensemble, we agree with the reviewer that this technique has been shown to benefit a model's performance.
> We also note that using an ensemble alone may not be considered a major novelty.
>
> Specifically, it does not offer a better way of harnessing the combined information from the query pair and the candidate target, while our second stage does.
>
> In contrast, our method explores the visiolingustic reasoning among the (reference image, text, candidate image) triplets through vision-and-language pre-trained models. We hope that our work provides valuable insight to future research on this task regarding better utilizing information within these entities.
>
> Finally, we note that our method is orthogonal to ensembles. It is possible to train multiple second-stage classifiers to form an ensemble, albeit at a *much* significant computational cost.
>
> > Component Originality: While the paper is easy to understand and well-written, it appears that most of its components are derived from existing methods. For example, the authors acknowledge that the objective is similar to and inspired by BLIP's ITM loss, raising questions about the novelty of the approach.
>
> We acknowledge the use of the vision-and-language pre-trained (VLP) network, BLIP.
> Meanwhile, regarding the image-text matching (ITM) loss, we note that it is a standard loss adopted for classification in vision-and-language tasks.
>
> We point out that it is common for recent work on downstream vision-and-language tasks to leverage a VLP network.
> The contribution, however, surrounds developing methods that solve the *specific* task, which requires insights.
>
> Here, we propose to explicitly reason over candidate images against the input (reference image, text)-pair, which is vastly different from existing work.
>
> Our significant performance increase comes from a novel dual-encoder design tailored for the input and output of composed image retrieval, which, we argue is valuable to future research.

---

### Review · Reviewer_iJWP · 2023-12-01

**Summary Of Contributions:**

This paper mainly targets composed image retrieval. The authors point out the problem of the previous methods that compare pre-computed image embeddings to the query text-engaged image embedding. To solve the problem they propose a two-stage pipeline that includes a candidate filtering stage, which filters out large proportion of candidate images, and a re-ranking stage which computes two candidate-aware embeddings with two separate branches. The proposed method is evaluated on two datasets including Fashion-IQ and CIRR, on which the results reflect the effectiveness of the method.

**Audience:**

Yes

**Broader Impact Concerns:**

None.

**Claims And Evidence:**

Yes

**Requested Changes:**

The authors can refer to the weaknesses for the problems that need to be solved. Moreover, I suggest the authors summarize the main notations in the overview in Sec.3 for better understanding.

**Strengths And Weaknesses:**

Strength:
1. The idea of leveraging two-stage pipeline is interesting.

2. The experiment results are strong.

Weaknesses:
1. As shown in the experiments, the proposed method is much slower than previous methods. While it is true that it significantly improves the performance, I am afraid that such a method would not be practical in real-life scenarios.

2. According to Sec. 4.1.3 the text encoder is optimized during training. In this way, how can the embedding $z_t$ be pre-computed?

3. It is not clear that how the model is optimized. Is the candidate filtering model used in the training of stage2 model? If so, how are they alternatively updated?

4. Ideally the encoder-t and encoder-$z_t$ should reflect the similarity between candidate image and $I_R$ and $t$ individually. I wonder if the authors can provide any experiments to explain the information learned in the dual encoder.

5. I wonder if it is necessary to conduct an experiment to show the sanity of candidate filtering. For example, what is the probability that the ground truth will be contained in the filtered set?

---

> ### Author Response · Authors · 2023-12-12
> **Response to review by reviewer iJWP**
>
> We thank reviewer iJWP for their reviews. Please see our responses below.
>
> > 1
>
> We acknowledge that our method is slower in inference.
>
> We point out that this work is not optimized for application scenarios where inference time is more crucial than retrieval accuracy. This is in line with most existing work on vision-and-language tasks, as discussed in Section 4.4.
>
> Ultimately, it is up to the user of the model to decide if the trade-off between performance and inference time is worth it.
>
> We note that real-life applications where data is processed offline do exist, in which case, users can afford the inference cost and will likely appreciate a significant improvement in retrieval accuracy.
>
> For instance, our method can be used to clean up the candidate image corpus (removing irrelevant candidates) for researchers who wish to construct future datasets on this task.
>
> > 2 and 3
>
> We would like to clarify our model pipeline.
> Specifically, the two stages are *not* trained jointly. Instead, they are two separate models.
>
> To this end, for the second stage re-ranking model, $z_t$ can be pre-computed because it is derived from the filtering model, which would have been trained during the first stage and is now frozen.
>
> As the two models are not trained simultaneously, no alternating weights update is involved.
>
> We are making changes to Figure 2 (model pipeline) to better clarify this by replacing the lines between the two stages with dashed lines.
>
> We will also make this point clear in the text.
>
> > 4
>
> The two encoders indeed focus on the two entities, $I_\text{R}$ and $t$, respectively.
> Our intention, however, is to merge the relevant information between $I_\text{R}$ and $t$, so that the merged embedding matches that of the target image.
>
> We have provided ablation studies on the effects of the encoders in Section 4.3.1 and Tables 3 and 4 (rows 1, and 2).
>
> Specifically, we tried removing either encoder-$t$ or encoder-$z_t$ and examined the performance impact, which confirms that the encoders play certain roles in the retrieval (please see discussions in Section 4.3.1).
>
> However, beyond that, it could be hard to explain the individual encoders because the information is shared through the transformer layers (see Figure 3 -- "Merge" module).
> To this end, the embeddings do not solely represent $I_\text{R}$ and $t$.
> Therefore, we are unable to isolate the embedding of either encoder and explore its relationship towards $I_\text{R}$ or $t$.
>
> > 5
>
> Please see the below tables.
>
> For Fashion-IQ, we report the ground truth coverage in detail for the three categories.
> For both tables, the $K$ ranges follow Tables 5 and 6.
>
> **GT Coverage** -- ground truth coverage, as in, the percentage value of queries with ground truth within the top-$K$.
>
> Please refer to Section 4.3.2 which discusses the effects of ground truth coverage. Tables 1 and 2 (captions) have already included the ground truth coverages of our choices of $K$, note that Table 2 is on CIRR test set, which is different than the validation set below.
> We will include further discussion and the below statistics in the form of line charts in the appendix.
>
> **FashionIQ**
>
> | K     | (GT Coverage perc) shirt | (GT Coverage perc) dress | (GT Coverage perc) toptee |
> | ----- | ------|-------|-------|
> | 10    | 45.04 | 43.78 | 49.62 |
> | 20    | 54.32 | 53.35 | 60.07 |
> | 30    | 59.96 | 59.20 | 65.48 |
> | 40    | 63.64 | 63.96 | 70.07 |
> | 50    | 67.47 | 67.38 | 72.62 |
> | 60    | 69.14 | 69.81 | 75.27 |
> | 70    | 70.80 | 71.99 | 77.15 |
> | 80    | 72.91 | 73.97 | 78.58 |
> | 90    | 74.63 | 75.90 | 79.76 |
> | **100**   | **75.86** | **77.24** | **81.18** |
> | 110   | 76.99 | 78.48 | 82.05 |
> | 120   | 77.82 | 79.47 | 83.02 |
> | 130   | 78.46 | 79.77 | 83.83 |
> | 140   | 79.20 | 80.66 | 84.96 |
> | 150   | 79.64 | 81.36 | 85.72 |
> | 160   | 80.18 | 82.15 | 86.13 |
> | 170   | 80.81 | 82.75 | 86.64 |
> | 180   | 81.26 | 83.59 | 87.15 |
> | 190   | 81.94 | 83.79 | 87.76 |
> | 200   | 82.43 | 84.38 | 88.32 |
>
> **CIRR validation set**
>
> | K   | Ground truth coverage (perc) |
> | --- | --------------------- |
> | 10  | 88.04                 |
> | 20  | 93.35                 |
> | 30  | 95.26                 |
> | 40  | 96.27                 |
> | **50**  | **97.08**      |
> | 60  | 97.51                 |
> | 70  | 97.90                 |
> | 80  | 98.16                 |
> | 90  | 98.52                 |
> | 100 | 98.68                 |
> | 110 | 98.83                 |
> | 120 | 98.92                 |
> | 130 | 99.07                 |
> | 140 | 99.14                 |
> | 150 | 99.23                 |
>
> Overall, we can observe a steep increase in the GT Coverage percentage value when $K$ is small. Meanwhile, the increments are smaller when $K$ is on the higher end, which is to be expected.
> This can be useful for determining the $K$ value of the re-ranking model -- i.e., a value that is neither too small nor too large.
>
> > Summarize the notations in the overview in Sec.3
>
> Thank you for the suggestion. We have made the necessary changes.

---

### Review · Reviewer_tcq4 · 2023-12-04

**Summary Of Contributions:**

This paper proposes a two-stage method to find an image that best matches a reference image and text pair. The first stage filters the large corpus to obtain hard negatives, while the second stage, a scoring task, employs a dual-encoder for re-ranking the hard negatives. This achieves a balance in inference cost via the pre-filtering. The effectiveness of the proposed method is validated on two datasets in different domains.

**Audience:**

Yes

**Claims And Evidence:**

Yes

**Requested Changes:**

1. Could you provide more experiments regarding my concern?
2. The choice of $K$ contributes to the inference time of the re-ranking model. Could you provide more information about the relationship between the number of hard negatives $K$ and the inference time? Additionally, how can we generally choose a suitable $K$?

Minor:

In the first paragraph of sec4.3.2：‘Table 5 and Table 6 studies’ <-study

**Strengths And Weaknesses:**

Strength:

1. Extensive experiments with detailed analysis of ablation studies are conducted.
2. Pre-filtering to balance the inference cost and performance.

Weakness / Concerns:

My primary concern revolves around the fact that this architecture, without incorporating supplementary structures for text, appears to solely compute text similarity. In simpler terms, I am apprehensive that this model might encounter difficulties with ambiguous sentences. For example, if we input an image of a yellow cat along with the text 'change to a white cat,' what outcomes could we anticipate?

---

> ### Author Response · Authors · 2023-12-12
> **Response to review by reviewer tcq4 (part-1)**
>
> We thank reviewer tcq4 for their review. Please see below for our response.
>
> > My primary concern revolves around the fact that this architecture, without incorporating supplementary structures for text, appears to solely compute text similarity.
>
> We are not entirely certain of the meaning of "solely compute text similarity", as our method harnesses information from the reference image as well (for instance, Figure 2 **right**: Encoder-$z_t$ takes in the embedding $z_t$, which contains information from the reference image $I_\text{R}$ on the left).
>
> Indeed, the task of composed image retrieval requires a model to take into consideration both the reference image and the text, so that they could be matched to a candidate target image.
>
> Ablation studies on image-only and text-only have been conducted in previous work. The results demonstrate the necessity of using both modalities of the input for this task. Please refer to Section 5.1 in [1].
>
> In addition, our ablation studies in Section 4.2.3 and Tables 3 and 4 (rows 1, and 2) also show similar outcomes. Specifically, we show that removing either encoder-$z_t$ or encoder-$t$ in the second stage yields subpar results. The former corresponds to a text-only experiment. The latter -- although not entirely image-only (as $z_t$ still contains text information) -- demonstrates that explicit information from text is crucial to the performance.
>
> Please feel free to follow up on our reply if it does not fully address your concern.
>
> > In simpler terms, I am apprehensive that this model might encounter difficulties with ambiguous sentences.
>
> On the topic of the text being ambiguous, we note that this is true for this task. Still, it remains a reasonable task to solve as user-provided text in real life can be concise, under-specified, and therefore, contains inherent ambiguities.
>
> A real-world system might mitigate this problem by, for instance, asking follow-up questions (as a dialogue system). Here, we restrict ourselves to the task at hand while following the existing evaluation protocols, which are not proposed by us.
>
> On the topic of evaluation protocols, we note that current standard ones are designed while recognizing the inherent ambiguity of the task, as they either use recall@10 and 50 (Fashion-IQ [2]) or recall@5 (CIRR [1]) instead of the much stricter accuracy (i.e., recall@1) -- even though each query only has *one* ground truth target labeled.
>
> > For example, if we input an image of a yellow cat along with the text 'change to a white cat,' what outcomes could we anticipate?
>
> For this example, we would anticipate the returned image to contain a white cat instead of a yellow cat, while being visually similar to the reference (input) image in all other aspects (e.g., the pose of the cats, the background, the viewing angle).
>
> Of course, such an image may not exist in the candidate image corpus. In reality, we would expect the returned image to be similar to what we have pictured above.
>
> A similar example, regarding the structure of the text, has been demonstrated in Figure 4 (b), where our model manages to retrieve the target image through the two-stage re-ranking.
>
> *To be continued to part-2 (characters limit)...*
>
> ---
>
> [1] Liu, Zheyuan, et al. "Image retrieval on real-life images with pre-trained vision-and-language models." Proceedings of the IEEE/CVF International Conference on Computer Vision. 2021.
>
> [2] Wu, Hui, et al. "Fashion iq: A new dataset towards retrieving images by natural language feedback." Proceedings of the IEEE/CVF Conference on computer vision and pattern recognition. 2021.

---

> > ### Author Response · Authors · 2023-12-12
> > **response to review by reviewer tcq4 (part-2)**
> >
> > *Continued from the above*
> >
> > > Could you provide more information about the relationship between the number of hard negatives $K$ and the inference time?
> >
> > The relationship between $K$ and the inference time is roughly linear (excluding fixed overheads such as model checkpoint loading, which takes only a small percentage of the total time).
> >
> > To validate this, please see the below tables on inference time vs. the choice $K$ for the re-ranking model, where our choices of $K$ in Tables 1 and 2 are **bolded**.
> > We are including the tables in the appendix.
> >
> > Each experiment is run independently three times on the same server. The results are then averaged.
> >
> > **CIRR**
> >
> > | K   | Inference time (sec) | Inference time (min:sec) |
> > | --- | -------------------- | ------------------------ |
> > | 10  | 316.15               | 05:16                    |
> > | 20  | 345.65               | 05:46                    |
> > | 30  | 378.36               | 06:18                    |
> > | 40  | 414.49               | 06:54                    |
> > | **50**  | **454.58**       | **07:35**                |
> > | 60  | 489.29               | 08:09                    |
> > | 70  | 526.06               | 08:46                    |
> > | 80  | 572.63               | 09:33                    |
> > | 90  | 605.35               | 10:05                    |
> > | 100 | 643.62               | 10:44                    |
> > | 110 | 682.38               | 11:22                    |
> > | 120 | 722.16               | 12:02                    |
> > | 130 | 761.04               | 12:41                    |
> > | 140 | 800.35               | 13:20                    |
> > | 150 | 836.07               | 13:56                    |
> >
> > **Fashion-IQ**
> >
> > | K   | Inference time (sec) | Inference time (min:sec) |
> > | --- | -------------------- | ------------------------ |
> > | 10  | 199.19               | 03:19                    |
> > | 20  | 228.10               | 03:48                    |
> > | 30  | 271.49               | 04:31                    |
> > | 40  | 320.96               | 05:21                    |
> > | 50  | 368.61               | 06:09                    |
> > | 60  | 413.76               | 06:54                    |
> > | 70  | 462.66               | 07:43                    |
> > | 80  | 514.90               | 08:35                    |
> > | 90  | 565.14               | 09:25                    |
> > | **100** | **612.75**       | **10:13**                |
> > | 110 | 663.41               | 11:03                    |
> > | 120 | 716.74               | 11:57                    |
> > | 130 | 768.27               | 12:48                    |
> > | 140 | 816.21               | 13:36                    |
> > | 150 | 866.60               | 14:27                    |
> > | 160 | 927.62               | 15:28                    |
> > | 170 | 977.03               | 16:17                    |
> > | 180 | 1029.20              | 17:09                    |
> > | 190 | 1079.83              | 18:00                    |
> > | 200 | 1138.57              | 18:59                    |
> >
> > We note that there is a slight discrepancy between the result above on Fashion-IQ with $K$=100 (approx. 10 minutes) and the inference time we noted in the paper ("around 9 minutes").
> >
> > We have updated the number in our paper by including these two tables in the appendix. We suspect the discrepancy is caused by the condition of our shared server at the time of running the experiments (e.g., if the system has any other loads).
> > We have taken special care when running the set of experiments in the above two tables to ensure that they are comparable to each other.
> >
> > We note that the difference does not affect our discussions in *Sec. 4.4 Inference time*.
> >
> > > Additionally, how can we generally choose a suitable $K$?
> >
> > The following discussion complements *Section 4.3.2 $K$ values in candidate re-ranking*:
> >
> > As discussed in this section, we note that a larger $K$ increases the ground truth coverage, but also leads to more negative candidates for the model to distinguish.
> > So a larger $K$ will increase the inference cost, but may not always improve the performance.
> >
> > In other words, there is an empirical upper bound for $K$, where setting $K$ beyond this upper bound yields no benefits in performance. This can be observed in Tables 5 and 6 for the two datasets respectively.
> >
> > When $K$ is below this empirical upper bound, there exists a trade-off between inference cost and performance.
> >
> > Ultimately, $K$ shall be empirically determined for the specific dataset and application, while considering the trade-off.
> >
> > > Minor (typo).
> >
> > Thank you for noting this typo, we have corrected this.

---

> > > ### Comment · Reviewer_tcq4 · 2024-01-14
> > > **Thank you for your response**
> > >
> > > Your efforts and time are appreciated. Most of my concerns have been addressed. I have no further concerns and recommend accepting your paper.

---

### Comment · Reviewer_zsrn · 2023-11-01
**Slow but reasonable approach**

This paper proposes to use candidate reranking methods to improve the performance of image retrieval systems.  Results show the proposed approach outperforms the state of the art methods including CLIP, BLIP and other methods. However, the high retrieval accuracy is obtained with a cost of a much slower retrieval speed (e.g., 9 minutes for the Fashion-IQ validation set. I think the proposed approach is reasonable -- its spirit is in line with the practice of Bert-based re-ranking strategy in web search. However, I also feel there are big rooms to reduce the computational cost and speed up the retrieval system.

---

> ### Author Response · Authors · 2023-12-12
> **Response to reviewer zsrn**
>
> We thank reviewer zsrn for their comments.
>
> As the topic of inference cost has also been mentioned by reviewers iJWP and HV9g, please refer to our response to them.

---

### Comment · Action_Editors · 2024-01-04
**Official Recommendation**

Dear reviewers,

Thanks for reviewing the paper. could you please submit the final score of this paper? thanks!

best,

AE

---

### Decision · Action_Editor_rHM3 · 2024-01-23

**Recommendation:** Accept as is

**Comment:**

This paper underwent a thorough review process involving multiple experts. Unfortunately, the reviewing process took longer than initially anticipated for certain reasons. Notably, two out of three reviewers express positive support for the paper. Overall, the authors delve into a compelling task and set a new benchmark for it. The majority of reviewers are in agreement that the paper's claims are substantiated by experimental results, a pivotal factor leading to the Assistant Editor's recommendation for acceptance.

 However, reviewers did raise concerns about the paper's perceived lack of novelty for ICLR and the efficiency of the proposed algorithm. These critiques are seen as valuable feedback, and addressing them could be a focus for future work.

**Audience:**

Re-ranking is an important technique for image retrieval tasks, which may be useful to the audience from Multimedia, Computer Vision and Machine Learning communities.

**Claims And Evidence:**

The authors looked into the difficulty of finding combined images, and they suggest that adding a second re-ranking step can really improve how well it works. The trick is to think of the re-ranking step as a scoring challenge. After testing it a lot, their method proves to be better.
At first, the reviewers were worried about how long it takes, what limits there are on the input, how the model is made better, and they wanted more experiments without certain parts. The authors mostly fixed all these worries the reviewers had.